# A gate tunable transmon qubit in planar Ge

Oliver Sagi[1] ✉, Alessandro Crippa [2], Marco Valentini[1], Marian Janik [1], Levon Baghumyan[1], Giorgio Fabris[1], Lucky Kapoor[1], Farid Hassani[1], Johannes Fink[1], Stefano Calcaterra [3], Daniel Chrastina [3], Giovanni Isella [3] & Georgios Katsaros [1]

Gate-tunable transmons (gatemons) employing semiconductor Josephson junctions have recently emerged as building blocks for hybrid quantum circuits. In this study, we present a gatemon fabricated in planar Germanium. We induce superconductivity in a two-dimensional hole gas by evaporating aluminum atop a thin spacer, which separates the superconductor from the Ge quantum well. The Josephson junction is then integrated into an Xmon circuit and capacitively coupled to a transmission line resonator. We showcase the qubit tunability in a broad frequency range with resonator and two-tone spectroscopy. Time-domain characterizations reveal energy relaxation and coherence times up to 75 ns. Our results, combined with the recent advances in the spin qubit field, pave the way towards novel hybrid and protected qubits in a group IV, CMOS-compatible material.

Hybrid quantum circuits interface different physical paradigms in the same platform, leading to devices with novel functionalities[1,2]. Specifically, hybrid qubits combining superconductors and semiconductors merge the maturity of superconducting quantum circuits with the inherent tunability of semiconductors. Gate-tunable transmons (gatemons)[3–5], parity-protected qubits[6], and Andreev spin qubits (ASQs)[7,8] are representative examples of recent achievements. In the development of hybrid qubits on novel material platforms, demonstrating coherent interaction between a microwave resonator and a Josephson junction is a paramount step[9–12]. Experimentally, the most direct approach to achieve this interaction is through capacitive coupling between a resonator and a gatemon circuit. This coupling enables dispersive readout[13] and, when combined with coherent control, serves as the foundation for progressing towards advanced hybrid circuits. A prominent example is ASQs, which combine the advantages of both superconducting and semiconductor spin qubit platforms[7,8,14–16].

Semiconductor spin qubits offer low footprint and high anharmonicity, but implementing long-range coupling over many qubits remains challenging, though remarkable advances have been recently demonstrated[17–19]. Superconducting transmon circuits provide well-mastered control and readout via microwave signals at the expense of large footprints and lower anharmonicity, posing challenges to scalability and fast operations[20]. Finding a suitable platform to merge the two systems is a formidable task. It requires a low microwave-loss

substrate (unless flip-chip technology is used[21]), transparent semiconductor-superconductor interfaces, and material free from nuclear spins for optimal spin qubit operation.

In the field of semiconductor spin qubits, Si is an attractive material choice as a result of the mature complementary metal-oxide-semiconductor technology[22] and isotopic purification[23]. Moreover, Si-based platforms have demonstrated the capability of hosting microwave resonators coupled to quantum dots[18,19,24–26]. However, the absence of Fermi level pinning makes it challenging to proximitize silicon without using doping or annealing techniques[27,28]. III–V semiconductor compounds, on the other hand, have emerged as the natural solution for hybrid devices due to the high-quality epitaxial aluminum (Al) growth, yielding a hard gap (i.e., free of subgap states) and a transparent interface[29–32]. Nevertheless, the non-zero nuclear spin limits the spin qubit coherence to ~10 ns[33].

Ge, and in particular Ge/SiGe heterostructures, have shown great potential for highly-coherent spin qubits and gate-tunable hybrid Josephson junctions[34–41], thus allowing the integration of semiconducting qubits with superconducting qubits on the same substrate.

Ge gatemons within Ge/Si core/shell nanowires have been recently realized[42,43]. However, while such CVD-grown core/shell wires have demonstrated ultrafast spin qubits[44,45] they face challenges in terms of spin dephasing times and scale-up. To circumvent both issues, we build on the success of two-dimensional implementations in

[1]Institute of Science and Technology Austria, Klosterneuburg, Austria. [2]NEST, Istituto Nanoscienze-CNR and Scuola Normale Superiore, Pisa, Italy. [3]L-NESS, Physics Department, Politecnico di Milano, Como, Italy. ✉ e-mail: oliver.sagi@ista.ac.at

InAs/InGaAs heterostructures[5,46] and realize a gatemon based on a Ge/SiGe heterostructure where superconductivity is induced into the Ge hole gas by proximity from an Al layer evaporated on top of the SiGe spacer. The gatemon resonant frequency is electrically tunable in a range of ~5 GHz and exhibits a quasi-monotonic, linear dependence on the gate voltage. The qubit relaxation time $T_1$ spans from ~80 ns to ~20 ns by varying the gate voltage, while $T_2^*$ does not show any clear trend. A control qubit on the same material stack using a superconductor-insulator-superconductor (SIS) junction exhibits a $T_1$ about five times longer than the semiconductor junction, and $T_2^*$ approaches $2T_1$. We discuss potential limitations in the Ge gatemon arising from the substrate and device layout.

## Results

### Device

An optical image of our device is shown in Fig. 1. We adopt a planar 'Xmon' geometry, known to maintain balance among coherence, connectivity, and swift control[47,48]. The core of the device is the semiconductor junction defined on the U-shaped mesa (see Fig. 1b, c) with a width of ~450 nm and a junction length of ~150 nm. The critical current $I_c$ through the semiconductor junction determines the Josephson energy ($E_J = \hbar I_c/2e$, with $\hbar$ the reduced Planck constant and $e$ the electron charge), which sets the qubit frequency together with the charging energy $E_c$ of the cross-shaped island. The island and the semiconductor junction together form the qubit circuit. The charging energy of the island is designed to be $E_c/h$ ~ 200 MHz to set the qubit frequency to be compatible with the 2–10 GHz range of our electronics and to operate in the transmon regime $E_J \gg E_C$[20]. The gatemon is coupled to a notch-type, $\lambda/4$ resonator with a loaded quality factor of $Q_l$ ~ 1400 (see Supplementary Fig. 1b). To measure the resonator, we

capacitively couple it to a 50 Ω coplanar waveguide transmission line, shown in purple in Fig. 1a and referred to as the feedline. The fabrication process flow is detailed in the Methods Section.

### Resonator spectroscopy

We investigate the gate tunable qubit-resonator interaction by monitoring the feedline scattering parameter $S_{21}$. The microwave power is kept low enough such that the average photon population of the resonator is approximately 1. By applying a voltage $V_{gate}$ on the Ti/Pd gate, we modify the critical current of the junction, which turns into a modulation of the qubit frequency: $f_{qubit} \propto \sqrt{I_c(V_{gate})}$. As displayed in Fig. 1e, the qubit exhibits one pronounced avoided crossing with the resonator. When the qubit is in resonance with the resonator, we observe two peaks in $|S_{21}|$ (at frequency $f_+$ and $f_-$, see Fig. 1f), a hallmark of two hybridized qubit-resonator states due to the strong coupling regime. To extract the coupling strength $g$ we fit the splitting: $\delta = f_+ - f_-$ according to the following equation[4]:

$$\delta = \sqrt{\left(f_{qubit} - f_c\right)^2 + 4\left(g/2\pi\right)^2},\qquad(1)$$

as illustrated in Supplementary Fig. 1c, where $f_{qubit}$ and $f_c$ are the bare qubit and resonator frequencies respectively. The fit yields $g/2\pi \approx 270$ MHz, consistent with electrostatic simulations. We also extract the qubit frequency:

$$f_{qubit} = (f_+ + f_-) - f_c,\qquad(2)$$

shown with a red dashed line in Fig. 1f. We obtain the bare resonator frequency from an independent high-power resonator spectroscopy

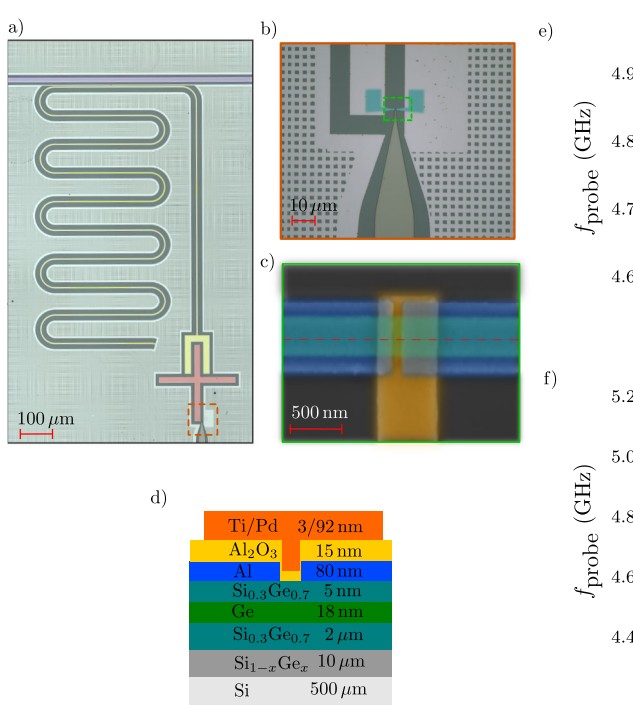

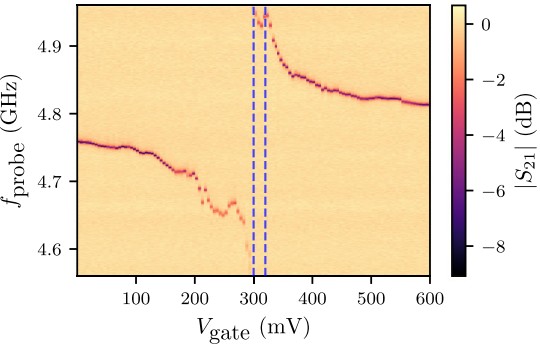

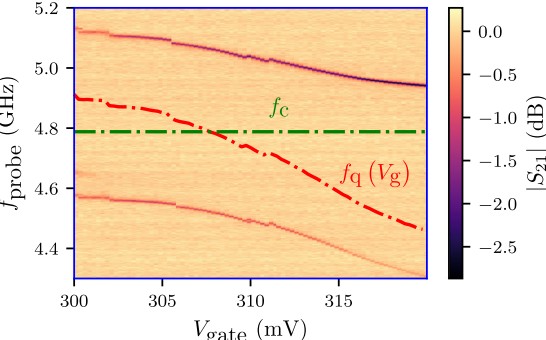

**Fig. 1 | Overview of the complete Ge gatemon device and resonator spectroscopy measured with a Vector Network Analyzer. a** Optical microscope image showcasing the entire device. A $\lambda/4$ notch-type coplanar waveguide resonator (yellow) is capacitively coupled to the cross-shaped qubit island (red) alongside a transmission line (purple) utilized for readout. The island is shunted to the ground through a gate-tunable semiconductor Josephson junction. **b** Close-up view of the U-shaped mesa (highlighted in light green). **c** False-colored SEM image of the junction, depicting the junction where the evaporated aluminum (blue) extends over the mesa (light green) to proximitize the whole mesa. The gate line (orange) is

intentionally extended on the right side to increase capacitance to the ground. **d** Cross-section of the wafer stack along the red-dashed line in Fig. 1c. **e** Gate-dependence of the normalized transmission through the feedline after background correction. The process of background correction is outlined in Supplementary Fig. 8 in the Supplementary Information. **f** Close-up view of the avoided crossing after background correction and boxcar averaging with a window size of 8 points. The dashed green line represents the bare resonator frequency, while the red dashed line indicates the qubit frequency extracted from the two peaks according to Eq. (2).

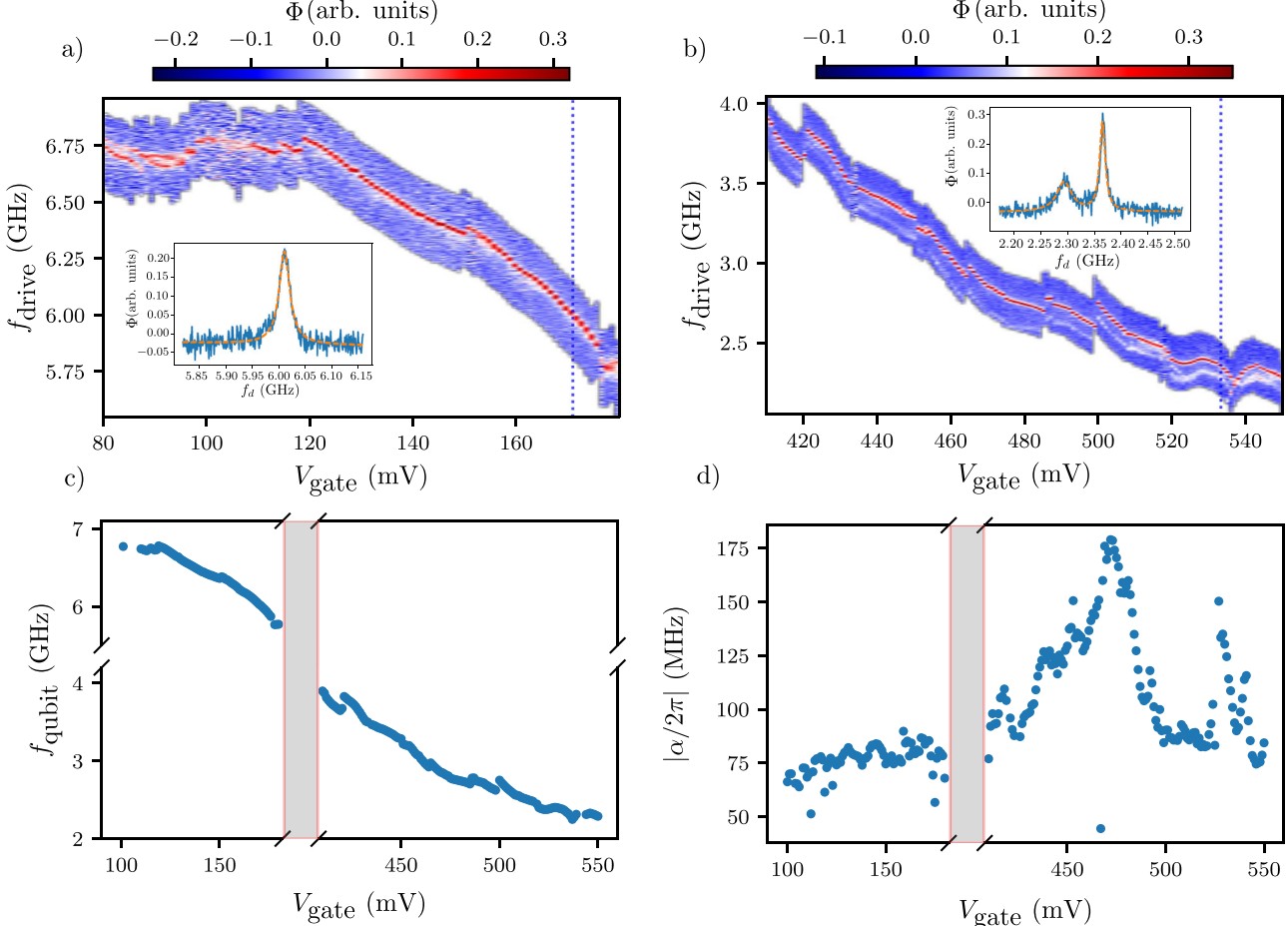

**Fig. 2 | Pulsed qubit spectroscopy. a, b** Two-tone spectroscopy data after subtracting the average of each column and normalizing the trace. In (**a**, **b**), the qubit frequency is set above (below) the resonator frequency. We measure the transmitted signal phase after a 2 μs excitation. In (**b**), we observe the $|1\rangle \rightarrow |2\rangle$ due to a residual excited state population. The insets show a linecut at $V_{gate} = 171$ mV and $V_{gate} = 533$ mV, respectively, along the blue dashed lines. The $x$ and $y$ axes

correspond to $f_{drive}$ and phase ($\Phi$), respectively. **c** Extracted qubit frequency from (**a**, **b**). We fit each trace with a Lorentzian (a pair of Lorentzian in the case of (**b**), and the center yields the qubit frequency. **d** Extracted anharmonicity from three-tone spectroscopy as explained in the main text. The anharmonicity is reduced compared to $-E_c$, indicating a non-sinusoidal current-phase relation.

measurement. We note that, in contrast to nanowire gatemons[6,49], we observe only one avoided crossing (refer to Supplementary Fig. 1 for a broader gate voltage range), suggesting a smooth and monotonic (albeit still suffering from discrete charge jumps) frequency dispersion. In Supplementary Fig. 4, we present additional resonator spectroscopy on a separate device with ~50 nm shorter channel length exhibiting similar behavior.

## Qubit spectroscopy

We now investigate the qubit frequency dependence in a broader gate voltage range than Fig. 1f. To do that, we move the qubit to the dispersive regime ($|\Delta| = |f_{qubit} - f_c| > g/2\pi$) to probe the qubit eigenstates. We adopt a conventional two-tone spectroscopy technique by applying a 2 μs long drive tone on the gate line followed by a 150 ns readout pulse on the resonator line. When the qubit drive matches $f_{qubit}$, we observe a peak in the resonator phase response. We choose the measurement frequency at each gate voltage to obtain all information in the phase of the measured signal. The observed peaks in Fig. 2a, b shift consistently with gate voltage; therefore, we attribute it to the qubit mode. The critical current and, thus, the qubit frequency increases with decreasing gate voltage since the charge carriers are holes. In Fig. 2b, a second, faint line appears, corresponding to the $|1\rangle \rightarrow |2\rangle$ transition, which is a signature of a residual thermal population of the excited state. We do not observe this transition when the qubit is

above the resonator. Similarly to the resonator spectroscopy, we see a monotonic gate voltage dependence of the qubit frequency interrupted by discontinuities at specific gate voltages, which we attribute to charge jumps. Below $V_{gate} < 100$ mV the qubit enters an unstable regime, exhibiting two peaks in two-tone spectroscopy likely due to a two-level fluctuator interacting directly with the qubit[50]. Another possible cause of the doubled line could be the unstable occupancy of highly transmitting Andreev levels in the junction due to quasiparticle poisoning, leading to fluctuations in $E_J$[51]. Therefore, we disregard the data at $V_{gate} < 100$ mV in the following analysis.

The qubit frequency can be tuned over a range spanning a few GHz, as shown in Fig. 2c. The junction is tunable even in a broader range, but the diminished readout visibility did not allow us to probe the qubit outside the 2–7 GHz range. The gray-shaded areas in Fig. 2c, d indicate the strong coupling regime discussed in Fig. 1f and parts of the dispersive regime where two-tone spectroscopy is not possible because of the reduced readout visibility.

We extract the anharmonicity at each gate voltage with a separate measurement by using three tones. A fixed frequency tone saturates the $|0\rangle \rightarrow |1\rangle$ transition while the frequency of a second tone is swept. We plot the measurement data in Supplementary Fig. 5, while in Fig. 2d, we summarize the extracted anharmonicities at all measured gate voltages. All measured anharmonicites are lower than $-E_c \approx -200$ MHz, a signature of non-sinusoidal current-phase relation[52],

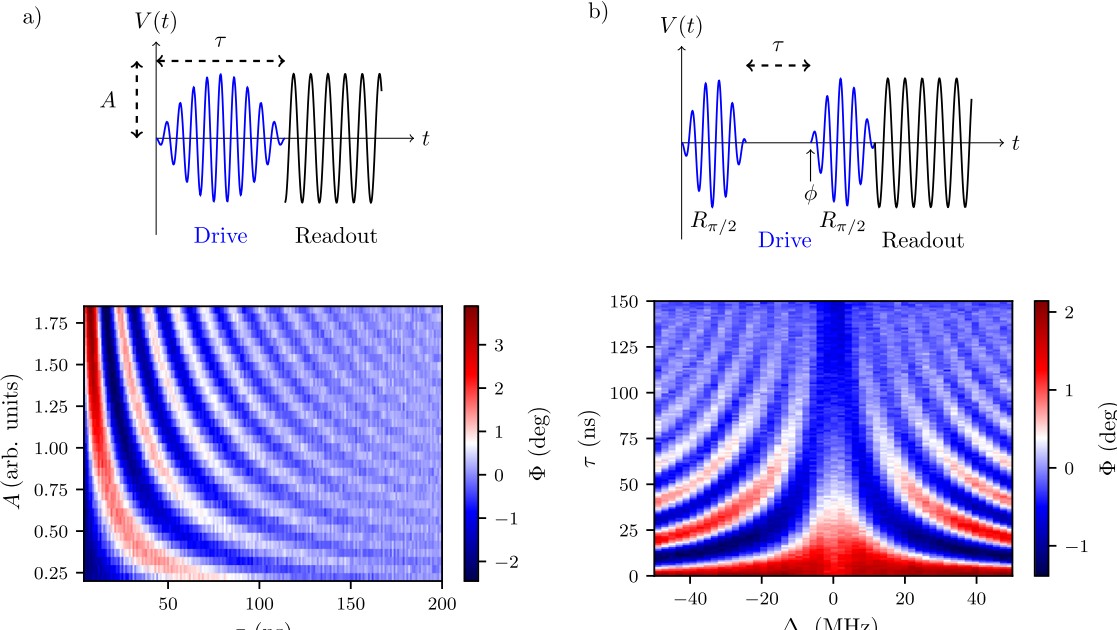

**Fig. 3 | Coherent control of the gatemon. a** Rabi oscillations at $f_{qubit} \approx 3.66$ GHz. We apply a cosine-shaped drive pulse directly on the gate line, followed by a readout pulse on the resonator line. **b** Ramsey fringes with virtual Z gates. We rotate the frame of reference to mimic an extra rotation. The detuning defines the angle of rotation: $\phi = \Delta_a \cdot \tau$. In both (**a**, **b**), the measured response averaged over 50000 traces, and the average of each column was subtracted to account for the slow drift of the readout resonator.

varying from −180 to −60 MHz. The variation in anharmonicity illustrates how the gate voltage influences transparency[53], with increased transparency resulting in lower anharmonicity, as seen in Fig. 2d. While reduced anharmonicity indicates a good interface quality between the proximitized Ge and the Ge forming the weak link, it also hinders fast qubit operations. In the case of a short, ballistic junction, a fourfold decrease in anharmonicity is expected compared to an SIS junction as the transparency reaches one[53]. Our device shows an approximately threefold decrease accompanied by pronounced fluctuations around certain gate voltage values. In Supplementary Fig. 4, a similar device with the same width but with a ≈50 nm shorter junction shows almost monotonic gate voltage dependence; the origin of this difference calls for further investigations.

### Coherent control
Next, we show coherent manipulation of the gatemon states. At a fixed gate voltage, we demonstrate *X-Y* rotations by applying a cosine-shaped drive pulse (indicated in blue in Fig. 3a) followed by a readout pulse on the resonator line. We dynamically vary the amplitude and the duration of the drive pulse at each sequence. The plot in Fig. 3a shows a typical Rabi pattern, featuring oscillations with an increasing frequency as the drive amplitude increases. Notably, the presented data is averaged over an hour, showcasing the stability of the sample at fixed gate voltage on that timescale.

After having calibrated the length of our $\pi$ pulse with the previously shown Rabi measurements, we move to *Z* rotations. In Fig. 3b, we employ virtual *Z* gates[54], involving two $\pi/2$ half pulses with an offset in the second pulse phase, introducing an artificial detuning ($\Delta_a$). This approach mitigates the limited visibility and unwanted AC Stark shift when using a detuned pulse. We obtain a Ramsey pattern by sweeping the virtual detuning and the idle between the two $\pi/2$ pulses.

### $T_1$ and $T_2^*$ measurements
Having calibrated the pulse sequences using previously conducted measurements, we proceed to characterize our qubit relaxation and coherence times. We determine the relaxation time $T_1$ by initializing the

qubit in the $|1\rangle$ state and adjusting the waiting time $\tau$ before applying the readout pulse. The characteristic decay time of the measured response is $T_1$, which is extracted by fitting the data with $A \exp(-\tau/T_1) + B$, as illustrated in Fig. 4a. Next, we measure the coherence time, $T_2^*$, using the same pulse sequence used in the Ramsey plot in Fig. 3b. We choose an artificial detuning greater than the decoherence rate to observe a sufficient number of oscillations. This allows us to reliably extract $T_2^*$ from $A \exp(-\tau/T_2^*) \cos(2\pi f\tau + \phi) + B + C\tau$, as depicted in Fig. 4b. Following this, we replicate the procedures mentioned above across multiple gate voltages, including qubit spectroscopy, Rabi, and Ramsey measurements. The results are presented in (c) and (d) of Fig. 4, where we display the extracted relaxation and coherence times as functions of the qubit frequency. The analysis yields a maximum $T_1$ of ≈73 ns across the scanned gate voltage range at $f_{qubit} \approx 2.8$ GHz. Additionally, we obtain a maximum $T_2^*$ of 71 ns within the explored gate voltage range.

### Al transmon on SiGe substrate
To gain deeper insights, we compare our device with a fixed-frequency reference transmon measured on a similar substrate and in the same setup, see Fig. 5a. This transmon shares identical capacitor geometry, and its junction size ( ̃200 nm × 180 nm) falls within the same range as the gatemon device presented ( ̃400 nm × 150 nm). The choice of a fixed-frequency transmon aims to minimize losses in relaxation channels other than dielectric losses in the substrate. Notably, the hole gas is removed on the entire chip, and the junction is of a conventional SIS type, with Al as superconductor (S) and AlO$_x$ as an insulator (I). We determine the qubit frequency and extract $T_1$ and $T_2^*$ times using techniques explained before. The measurements are detailed in Fig. 5b–d. We find $T_1 \approx 109$ ns and $T_2^* \approx 196$ ns, very close to the upper limit of $2T_1$.

### Discussion
To elucidate the underlying loss mechanisms of our gatemon, we draw comparisons between our findings on the semiconductor transmon and analysis from bare resonators measured on the same substrate, as outlined in ref. 40. We then complement the discussion by including the Al transmon figures in the comparison (Fig. 5).

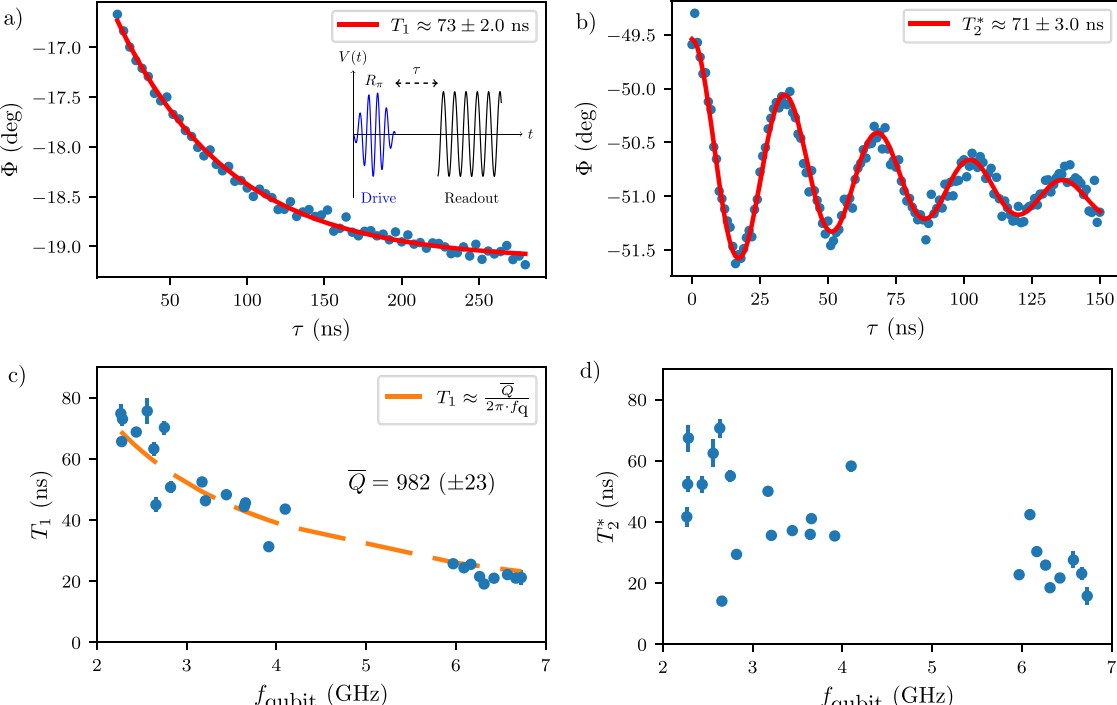

**Fig. 4 | Relaxation and coherence time measurements.** Measurements depicted in panels (**a**, **b**) were performed at different readout frequencies. **a** $T_1$ measurement at $f_{qubit} \approx 2.8$ GHz. A 10 ns $\pi$ pulse brings the qubit to the excited state, followed by a certain waiting time and the readout pulse. The solid red line is a fit to the exponential curve. **b** $T_2^*$ measurement at $f_{qubit} \approx 2.8$ GHz. The pulse sequence is identical to the one in Fig. 3b. The solid curve fits a damped sinusoidal curve on a linear background. **c** Relaxation time measurements as a function of qubit frequency. The error bars represent the standard deviations of the fit depicted above. The dashed line indicates a fit to the function shown in the legend. We extract an effective quality factor lower than the $Q_i$s of bare resonators on a similar substrate. **d** Coherence time as a function of qubit frequency. The measured coherence times do not reach the $2T_1$ limit. The error bars represent the standard deviations of the fit.

We start by converting the measured relaxation times into quality factors by using the expression $T_1 = \overline{Q}/(2\pi f_{qubit})$[55]. In Fig. 4c, we fit the obtained relaxation times of the gatemon with this equation, which yields an "average" quality factor $\overline{Q}$ nearly five times lower than that measured with bare resonators on the same substrates[40]. This suggests additional losses beyond those originating from the substrate. This conclusion is further supported by the fact that the Al transmon $T_1$ is approximately five times higher than in the semiconductor device at the same qubit frequency. We then delve into the disparities between the semiconductor gatemon and the Al transmon/resonators to shed light on the potential loss channels.

The first significant distinction is the gatemon mesa structure, which features a conducting two-dimensional hole gas covered with Al. Though a conductive layer normally would cause extra dissipation, ref. 40 demonstrated that proximitized germanium (highlighted in red in Fig. 1b) does not introduce significant additional losses. Resonators defined atop proximitized germanium showed comparable internal quality factors (within a factor of two) to those defined with the quantum well etched away. Therefore, the presence of proximitized germanium alone cannot solely account for the reduced energy relaxation time.

The gatemon differs notably from both the SIS transmon and bare resonators due to the presence of the Pd gate, which may introduce dissipative losses in the gatemon architecture due to its normal metal properties[56]. On top, the gate itself functions as a coplanar waveguide transmission line, adding another relaxation channel, but we rule out relaxation to the gate line or resonator as a limiting factor (refer to the Supplementary Information for further details). Moreover, Atomic Layer Deposition grown $AlO_x$ is deposited across the entire gatemon device, including the qubit capacitor area where the electric field is concentrated, which can lead to additional dielectric losses[46]. It is also worth mentioning that semiconductor junctions may exhibit subgap states, potentially leading to quasiparticle losses due to external radiation[49,57]. We highlight that a transmon on this substrate is expected to be limited by the same dielectric losses as the bare resonator with an internal quality factor of ~5000, explaining the observed relaxation times of ~100 ns. Finally, the analysis of coherence times of Fig. 4d reveals a non-monotonic trend, diverging from the $2T_1$ limit. This observation indicates that the coherence of the qubit is not fully constrained by energy relaxation but also influenced by dephasing attributed to charge fluctuators within the oxide (covering the entire chip)[5].

In summary, we have demonstrated a gate-tunable superconductor-semiconductor qubit on planar Ge. Our qubit shows broadband tunability and quasi-linear frequency dispersion, rendering it a promising element for ASQs and superconducting circuits requiring tunable elements. We have characterized the energy relaxation and coherence times, pinpointing the possible limitations with identified areas for improvement. One natural upgrade is to replace the Ti/Pd with either Al, Nb, or Ta, as exemplified in ref. 56. Losses in the oxide can be mitigated by lifting or etching the oxide with an additional fabrication step or switching to a low-loss dielectric, e.g., hBN[58]. To prolong the lifetimes of our gatemons beyond the constraints imposed by dielectric losses in the buffer, one can explore strategies such as deep reactive ion etching[59] and flip-chip technology[21,60].

With germanium already having solidified its position as a prominent player in the spin qubit domain[37,61,62], it is naturally considered an alternative platform for ASQs since current experiments are limited by dephasing due to nuclear spins[15]. To compete with III–V systems in ASQ platforms, the potential of Ge still needs to be demonstrated in the field of superconducting qubits. In that context, our work constitutes the entry of planar Ge in the field of hybrid qubits. Integrating

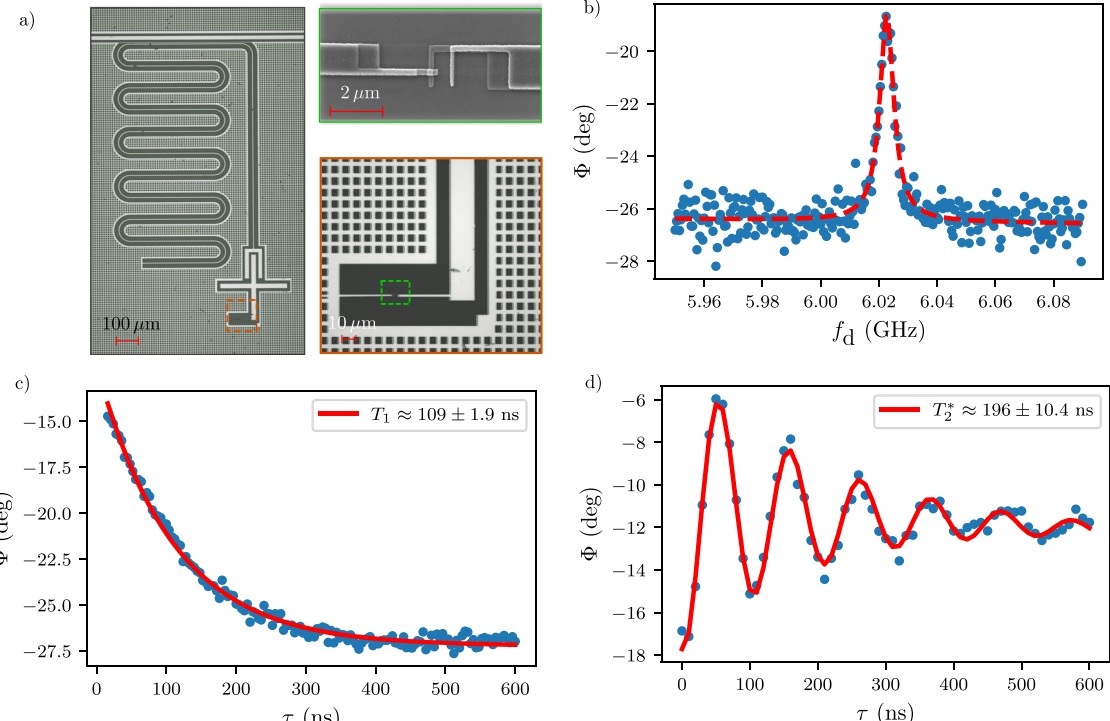

**Fig. 5 | Al transmon characterization on etched SiGe substrate. a** Overview of the device. We replaced the semiconductor junction with a shadow-evaporated Al-AlOₓ-Al junction. Otherwise, we kept the device geometry the same. **b** Qubit characterization with pulsed two-tone spectroscopy. The red dashed line represents a fit to a Lorentzian function. We choose the measurement frequency such that all information is contained in the phase of the measured signal. **c** Relaxation time measurement using the same pulse sequence shown in Fig. 4. The solid red line is a fit to an exponential decay. **d** Coherence time measurement using the same pulse sequence as shown in Fig. 4. The solid red line is a fit to a decaying sinusoidal oscillation.

gatemons in ASQ circuits will loosen up readout requirements by allowing in-situ frequency tuning and capacitive coupling to resonators[63]. Future challenges will involve addressing issues introduced by the shallow quantum wells[40] and improving the microwave properties of the substrates[64]. Very recently, a proximitized quantum dot in Germanium was demonstrated, further highlighting the increasing prominence of Ge in the field of hybrid quantum systems[65].

## Methods

### Sample fabrication

First, we define the mesa structure with a depth of ~60 nm by etching away the hole gas with an $SF_6$-$O_2$-$CHF_3$ reactive ion etching process. The hole gas is etched away everywhere, except the mesa area, to remove the conducting hole gas below the microwave circuit. The etching depth ( ~60 nm) was selected to ensure reliable removal of the quantum well, accounting for fluctuations in the etching rate. Additionally, this depth allows the gate to be deposited with a thickness compatible with our lift-off process and guarantees continuity when the metal climbs the mesa. The Josephson junctions (JJs) are formed by evaporating at room temperature 80 nm Al on the mesa after a 15 s buffered HF dip. The Al is oxidized for 2 min at 10 mbar pressure before venting the evaporation chamber. The microwave circuitry is defined simultaneously with the JJ to reduce the number of fabrication steps. Finally, a plasma-assisted aluminum oxide, approximately 15 nm thick, is deposited at 150 °C, followed by the evaporation of Ti/Pd gates, where Ti was used as an adhesion layer. The thickness of the Al and Pd were chosen to be 20 nm greater than the mesa depth to ensure proper climbing of the edge.

### Measurements

We performed all measurements in a cryogen-free dilution refrigerator with a base temperature below 10 mK. The sample was mounted on a custom printed circuit board thermally anchored to the mixing chamber of the cryostat, and electrical connections were made via wire bonding. The schematic of our measurement setup is shown in Supplementary Fig. 10.

## Data availability

All experimental data included in this work will be available at the ISTA Research Data Repository[66].

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

## Acknowledgements

We acknowledge Lucas Casparis, Jeroen Danon, Valla Fatemi, Morten Kjaergard and Javad Shabani for their valuable insights and comments. We thank Simon Robson for proofreading the manuscript. This research was supported by the Scientific Service Units of ISTA through resources provided by the MIBA Machine Shop and the Nanofabrication facility. This research was funded in whole or in part by the Austrian Science Fund (FWF) [https://doi.org/10.55776/I5060 and https://doi.org/10.55776/P36507]. We acknowledge financial support from the Horizon Europe Framework Program of the European Commission through the European Innovation Council Pathfinder grant no. 101115315 (QuKiT). For Open Access purposes, the author has applied a CC BY public copyright license to any author accepted manuscript version arising from this submission. We also acknowledge the support of the NOMIS Foundation, the European Union's Horizon 2020 research and innovation program under Grant Agreement No 862046 and the NextGenerationEU PRIN project 2022A8CJP3 (GAMESQUAD) for partial financial support.

## Author contributions

O.S. and L.B. were responsible for fabricating the devices, with O.S. conducting measurements and analyzing the data under the guidance of G.K. M.V. and O.S. collaborated on developing the nanofabrication recipe for hybrid devices, while M.J. and O.S. were involved in developing the microwave technology for the Ge/SiGe heterostructures. G.F. contributed to device fabrication and measurements. S.C., D.C. and G.I. were responsible for the growth of the Ge quantum well. F.H. and L.K. provided assistance during the experiments. F.H., L.K. and J.F. provided the transmon nanofabrication recipe. O.S. A.C. and G.K. discussed the results and wrote the manuscript with contributions and feedback from all co-authors.

## Competing interests

The authors declare no competing interests.
