## [Peer Review File · Nature Communications]

REVIEWERS' COMMENTS

Reviewer #1 (Remarks to the Author):

This paper reports on the implementation and measurement of a gatemon qubit in planar Ge. The qubit is characterized using resonator and two-tone spectroscopy, and shown to be electrically tunable in a frequency range of ~ 5 GHz. Coherent control of the qubit is demonstrated, and qubit relaxation and coherence times are characterized. A remarkable and in my mind for the first time, an Al transmon qubit is fabricated on the same substrate, in order to draw a direct comparison, which is used to determine the loss mechanisms.

Overall, this work is an important contribution to the field, and the manuscript is quite clear and well written. I would recommend publication.

Please consider my minor comments below:

- color coding in Fig. 1 a)-d) is a bit confusing. At first I was assuming that all color coding would be consistent with the stack in Fig. 1)d), but from the text I see that this is not the case, and that red in panel b represents proximitized germanium, and seems to change color in panel c) where it becomes blue Al on green mesa, instead of red as it was in b). I think that the clarity of this figure would benefit from having the colors standardized, and clarified.

- In Fig. 4, the caption for panel b reads 'T₂ measurement at $f_q \approx 2.8$ GHz. The pulse sequence is identical to the one in Fig. 3b. The solid curve fits a damped sinusoidal curve on a linear background. The error bars represent the standard deviations of the fit.'

As far as I can see, there are no error bars in panel b. Was this meant for a different part of the caption?

- Another minor point: Fig. 4 d does not have the 'd)' label in the figure.

- Could you elaborate, either in the main text of the methods section, on the choice to etch the mesa structure with a depth of ~ 60 nm? Is the structure still conducting at this depth?

- The manuscript is quite focused on the comparison between the gatemon and transmon measured here on the same substrate, with the takeaway that the transmon is less lossy than the gatemon. This is indeed a valuable comparison. I would suggest to mention that overall low coherence times of both transmon and gatemon are due to low internal Q/substrate? For example, it is expected that a transmon on this substrate would have a T₁ of only 100 ns due to... This would more clearly show a future path for improvement.

Reviewer #2 (Remarks to the Author):

The authors of this paper report measurements of a transmon using a planar Ge hole gas. They demonstrate gate control of the transmon frequency and also measure the coherence times of their qubit. This result is interesting for the Andreev spin qubit community since in the usual platform of InAs, the coherence times of the ASQs are thought to be limited by nuclear spins. Germanium, in contrast, is expected to be free of nuclear spins and is a promising platform for ASQ implementation. This paper takes the first step in building coherent superconducting devices on planar Ge though the very low lifetimes (below that of current ASQs in InAs) of the implemented transmon limit widespread utility. The paper is written clearly and the measurements are sound. I would like the authors to provide some additional information to better understand what is limiting their platform:

1. What is the measurement temperature of the dilution refrigerator and the transmon temperature? The authors state that they observe transitions to higher-excited state due to residual thermal population but it would be useful to know what temperature or percentage in the excited state this corresponds to in the transmon.
2. The noise spectral density that affects the T_1 of the measured transmons should depend on temperature as well. In Fig. 4c, what is the assumed temperature for this fit to extract the $Q \sim 1000$?

Reviewer #3 (Remarks to the Author):

The manuscript titled "A gate tunable transmon qubit in planar Ge" (NCOMMS-24-18108), by O. Sagi and the rest of G. Katsaro's group, builds upon results by the same group, who previously investigated the physics of Josephson junctions (JJs) and superconducting resonators implemented on germanium quantum wells (Ge QW) [Ref. [39], published in Nature Communications]. In the present work, the authors combine these two previously established building blocks to engineer a hybrid gatemon qubit where the JJ is implemented on a superconductor-proximitized Ge QW. As I elaborate on below, I am certain that the manuscript would be of great interest to the broad readership of Nature Communications and, thus, I fully support its publication as an article.

I believe that these results are a great addition to the nascent field of Andreev spin qubits (ASQ). To date, ASQs have been successfully implemented in InAs nanowires, but have shown reduced coherence times due to hyperfine interaction between the qubit spin and the bath of nuclear spins in InAs. For this reason, there is increasing interest in the community towards switching to a new material platform. Such material should (1) be a semiconductor (2) where quantum dots can be defined, (3) that can be proximitized with a superconductor, (4) that has a reduced presence of nuclear spins, (5) that shows a strong spin-orbit interaction and, importantly, (6) should be compatible with the implementation superconducting resonators and transmons for readout.

Germanium has raised a strong interest in the ASQ community, as it has proven to fulfill the first 5 points. While alternative work on flip-chip implementations for readout exists (see e.g. Ref. [20]), it's not directly clear how such geometries could be used to readout combinations of ASQs coupled in parallel. The present manuscript shows the first realization of a resonator+transmon circuit directly integrated on a stack with a Ge QW and thus proves that germanium fulfills all main requirements listed above. For this reason, I anticipate that this paper will be a stepping stone for the realization of germanium ASQs.

I find that the manuscript is beautifully written. The content is complete and detailed while being to the point, making it easy to follow. The text has a good structure, with a suitable motivation, a clear presentation of the results, a complete discussion and accurate conclusions. Moreover, I found the supplementary explanation of the various possible loss mechanisms very insightful. Finally, I'd like to highlight that the manuscript presents data from two different devices. This is not always the case in the field of hybrid superconducting-semiconducting devices, due to the high complexity of the device fabrication and of the measurements performed.

Below, I list some concrete comments and questions for the authors:

Minor questions:

- In the caption to Fig. 1 the authors mention that "The gate line (orange) is intentionally extended on the right side to increase capacitance to the ground." Why is that?
- Do the authors know whether the gate jumps are reproducible or not? If they know, it'd be great if this information was in the manuscript.
- In Fig. 2b the authors say that the second visible transmon frequency corresponds to f_{12} of the transmon. In one of the (continuously driven) supplementary figures, they instead mention that they observe $f_{02}/2$. What are the arguments for these two different identifications?

- In line 173, the authors suggest a “two-level fluctuator interacting directly with the qubit” as the origin for the doubled line in two-tone at high E_J . Could they elaborate on exactly how the TLS affects the qubit frequency? Is it a charge TLS affecting the E_J value?
- Related to the point above, the doubled-line behavior is correlated with the gate region of high transparency (given the anharmonicity measurement). Therefore, an alternative explanation for the doubled line would be the presence of a highly transmissive Andreev level in the junction. If the occupancy of such Andreev level changes over time due to quasiparticle poisoning, the effective E_J in turn takes two different values. Why do the authors rule out this possible origin?
- Why is there a mismatch between y axes in Fig. 4a and Fig. 4b?
- In line 319, the authors say that potential subgap states could lead “to quasiparticle losses due to external radiation”. It’s unclear to me how the subgap states contribute to the quasiparticle losses. Could the authors elaborate?

Minor comments:

- In the caption to Fig. 1 the authors mention that they plot the “normalized resonator transmission after a background correction”. While clear to the experts in the field, I believe that “resonator transmission” is not completely accurate. I suggest saying “transmission through the feedline” or anything equivalent instead. Also, it would be great if it was clearly stated what the background correction consists of exactly.
- In line 117, the authors refer to the feedline which, while visible in Fig. 1, hasn’t actually been introduced in the main text. I find that the manuscript would benefit from briefly introducing this object in the device section before referring to it.
- In the caption to the insets in Fig. 2a and b, I don’t find the comment explaining what the x and y axes are needed.
- Should the approximate sign in the legend label in Fig. 4c be an equal sign?
- I believe there is a typo in line 774: impedance -> capacitance
- I find it misleading to use the word “feedline” in line 776, as there is another feedline in the chip. It could instead be called "driveline" or "gate line" here.
- There are multiple symbol inconsistencies that, while not affecting the understanding of the scientific results, are detrimental to the smooth reading of the text. The manuscript would benefit from a more careful use of symbols. Some examples are:
 - o $V_{\{\text{rm gate}\}}$ and $V_{\{\text{rm g}\}}$ are used to refer to the same variable.
 - o Same for $I_{\{\text{rm c}\}}$ and I_c .

- o Same for E_J and E_{J} .
- o Same for E_C and E_{c} .
- o Same for f_d , f_{d} and f_{drive} .
- o Same for f_q , f_{q} and f_{qubit} .
- o Same for Z , Z_0 and R in the Supplement.
- o The symbol Δ is being used to denote two different things throughout the manuscript (the resonator–qubit detuning and the artificial (effective) Ramsey detuning). I suggest using two different symbols.

Reference suggestions:

While the manuscript references the previous literature appropriately and in a complete manner, I do have some minor bibliography comments and suggestions for the authors to consider:

1. I suggest that the authors cite Koch2007 (currently Ref. [48]) when the transmon qubit is introduced in line 44.
2. The authors could consider adding a reference to [van den Berg et al., PRL (2013)] next to their reference to Nadj-Perge2010 (currently Ref. [32]) in line 57. Both works investigate the spin coherence in III-V materials and find the T_2 times to be in the ns range.
3. I find that the recent preprint arXiv.2405.02013 beautifully complements this work when thinking about ASQ applications. The authors could consider adding a reference to this preprint in their discussion on the benefits of Ge as an ASQ platform.
4. I believe there is a typo in the reference in line 342. An alternative to the current reference would be [A. Bruno et al., Appl. Phys. Lett. 106, 182601 (2015)], but the authors might have had a different option in mind.

Signed: Marta Pita-Vidal

Reply to Referee report for the manuscript “Gate-tunable transmon in planar Ge”

(Dated: July 4, 2024)

I. REPLY TO REFEREE 1

We thank the referee for the positive remarks and the comments on the paper.

- color coding in Fig. 1 a)-d) is a bit confusing. At first I was assuming that all color coding would be consistent with the stack in Fig. 1d), but from the text I see that this is not the case, and that red in panel b represents proximitized germanium, and seems to change color in panel c) where it becomes blue Al on green mesa, instead of red as it was in b). I think that the clarity of this figure would benefit from having the colors standardized, and clarified.

We agree with the referee that an inconsistency in colour coding may mislead the reader. Therefore, we have changed the colour of the mesa in Fig. 1b to light green, in agreement with Fig. 1c. We have also modified the colour of the transmission line to purple to avoid a double use of blue.

- In Fig. 4, the caption for panel b reads T2 measurement at $f_q \approx 2.8$ GHz. The pulse sequence is identical to the one in Fig. 3b. The solid curve fits a damped sinusoidal curve on a linear background. The error bars represent the standard deviations of the fit. As far as I can see, there are no error bars in panel b. Was this meant for a different part of the caption?

The last sentence of the caption was indeed meant for panel d. We thank the referee for spotting this unintentional mistake.

- Another minor point: Fig. 4 d does not have the ‘d)’ label in the figure.

We have added the label ‘d)’, as pointed out.

- Could you elaborate, either in the main text of the methods section, on the choice to etch the mesa structure with a depth of ~ 60 nm? Is the structure still conducting at this depth

We have complemented the Methods section with the following sentences:

The etching depth (~ 60 nm) was selected to ensure reliable removal of the quantum well, accounting for fluctuations in the etching rate. Additionally, this depth allows the gate to be deposited with a thickness compatible with our lift-off process and guarantees continuity when the metal climbs the mesa.

- The manuscript is quite focused on the comparison between the gatemon and transmon measured here on the same substrate, with the takeaway that the transmon is less lossy than the gatemon. This is indeed a valuable comparison. I would suggest to mention that overall low coherence times of both transmon and gatemon are due to low internal Q/substrate? For example, it is expected than a transmon on this substrate would have a T1 of only 100 ns due to... This would more clearly show a future path for improvement.

We have added an extra sentence in the Discussion section to emphasize the substrate losses once more:

We highlight that a transmon on this substrate is expected to be limited by the same dielectric losses as the bare resonator with an internal quality factor of ~ 5000 , thus explaining the observed relaxation times of ~ 100 ns.

II. REPLY TO REFEREE 2

We thank the referee for appreciating our work.

- What is the measurement temperature of the dilution refrigerator and the transmon temperature? The authors state that they observe transitions to higher-excited state due to residual thermal population but it would be useful to know what temperature or percentage in the excited state this corresponds to in the transmon.

We agree with the referee that the effective qubit temperature is a crucial metric for benchmarking qubit systems. Accurate determination of qubit temperature relies on a high-fidelity state assignment, which necessitates single-shot readout [1, 2]. Unfortunately, our readout parameters deviated significantly from the ideal condition of ($2\chi = \kappa$ and $t_{\text{meas}} \ll T_1$) [3], implying a deteriorated SNR which has rendered state assignment in a single measurement unattainable. Since the relaxation time was comparable to the measurement time, the visibility was significantly reduced. To address this, we performed extensive averaging (50,000 in time-domain measurements) to achieve a satisfactory SNR. Moving forward, we aim to enhance the relaxation time of our gatemon and incorporate a parametric amplifier into our measurement setup to resolve this issue.

The measurement temperature of the mixing chamber plate of the dilution refrigerator was ~ 10 mK.

- The noise spectral density that affects the T_1 of the measured transmons should depend on temperature as well. In Fig. 4c, what is the assumed temperature for this fit to extract the $Q \sim 1000$?

We agree with the referee that in a quantum mechanical treatment, utilizing Fermi’s golden rule, the estimated decay rates depend on the effective temperature [4]. Since the transmon is essentially an anharmonic oscillator, it can be well compared to a resonator (harmonic oscillator) characterized by a quality factor. By following analyses in previous works [5–8], we have used an ‘effective’ quality factor derived from the relaxation time: $Q = T_1 \cdot 2\pi f_q$, encompassing all the microscopic details.

III. REPLY TO REFEREE 3

We thank Referee 3 for the careful analysis of the paper and the points that arose.

- In the caption to Fig.1 the authors mention that “The gate line (orange) is intentionally extended on the right side to increase capacitance to the ground.” Why is that?

We wanted to increase the capacitance of the gate line to the ground to suppress relaxation to the gate, as in Ref. [9].

- Do the authors know whether the gate jumps are reproducible or not? If they know, it’d be great if this information was in the manuscript.

We observed that the majority of the gate jumps occur at the same gate voltage. We have added extra gate sweeps, shown in Supplementary Fig. 9, with two qubit spectroscopy measurements with 1 day difference.

- In Fig. 2b the authors say that the second visible transmon frequency corresponds to f_{12} of the transmon. In one of the (continuously driven) supplementary figures, they instead mention that they observe $f_{02}/2$. What are the arguments for these two different identifications?

The arguments are mainly technical. In the main text, we extracted the anharmonicity with pulsed measurements using three tones. We opted for pulsed measurements to eliminate the effect of AC stark shift and dephasing due to the photon shot noise in the resonator. For the sake of time, the sample shown in the supplementary was characterized with continuous-wave measurements that yield the desired information with less optimization.

- In line 173, the authors suggest a “two-level fluctuator interacting directly with the qubit” as the origin for the doubled line in two-tone at high E_J . Could they elaborate on exactly how the TLS affects the qubit frequency? Is it a charge TLS affecting the EJ value?

In superconducting circuits, defects, impurities, or trapped charges can create parasitic quantum two-level systems (TLS) interacting with oscillating electric fields via their electric dipole moments. Due to the random structure of materials, TLS resonance frequencies can vary widely, and those near the qubit resonance can significantly affect qubit energy relaxation and frequencies [10]. If a TLS frequency is sufficiently close to the qubit frequency, their coherent interaction can lead to vacuum Rabi splitting, which appears as a double qubit line.

- Related to the point above, the doubled-line behavior is correlated with the gate region of high transparency (given the anharmonicity measurement). Therefore, an alternative explanation for the doubled line would be

the presence of a highly transmissive Andreev level in the junction. If the occupancy of such Andreev level changes over time due to quasiparticle poisoning, the effective E_J in turn takes two different values. Why do the authors rule out this possible origin?

We thank the referee for the insightful remark. We have added such an alternative explanation in the main text in the following form:

Another possible cause of the doubled line could be the unstable occupancy of highly transmitting Andreev levels in the junction due to quasiparticle poisoning, leading to fluctuations in E_J .

- Why is there a mismatch between y axes in Fig. 4a and Fig. 4b?

The two measurements were performed at different readout frequencies. We have now added extra clarification to the caption.

- In line 319, the authors say that potential subgap states could lead “to quasiparticle losses due to external radiation”. It’s unclear to me how the subgap states contribute to the quasiparticle losses. Could the authors elaborate?

The presence of subgap states leads to an effective reduction of Δ^* , which increases the quasiparticle-induced decay rate, which scales as $\exp(-\Delta^*/k_B T)$ [11].

- In the caption to Fig. 1 the authors mention that they plot the “normalized resonator transmission after a background correction”. While clear to the experts in the field, I believe that “resonator transmission” is not completely accurate. I suggest saying “transmission through the feedline” or anything equivalent instead. Also, it would be great if it was clearly stated what the background correction consists of exactly.

We added an extra figure to the Supplementary Information section. The background correction process is shown in Supplementary Fig. 8.

- In line 117, the authors refer to the feedline which, while visible in Fig. 1, hasn’t actually been introduced in the main text. I find that the manuscript would benefit from briefly introducing this object in the device section before referring to it.

We have inserted an extra sentence in the device section:

To measure the resonator, we capacitively couple it to a 50 Ω coplanar waveguide transmission line, shown in purple in Fig.1a and referred to as the feedline.

- In the caption to the insets in Fig. 2a and b, I don’t find the comment explaining what the x and y axes are needed.

We have slightly modified the sentence in the caption. It now reads:

The insets show a linecut at $V_{gate} = 171$ mV and $V_{gate} = 533$ mV, respectively, along the blue dashed lines.

- Should the approximate sign in the legend label in Fig. 4c be an equal sign?

We have replaced the approximate sign with an equal sign.

- I believe there is a typo in line 774: impedance \rightarrow capacitance

We have corrected the typo.

- find it misleading to use the word “feedline” in line 776, as there is another feedline in the chip. It could instead be called “driveline” or “gate line” here.

As suggested, we have replaced ‘feedline’ with ‘drive line’.

- I suggest that the authors cite Koch2007 (currently Ref. [48]) when the transmon qubit is introduced in line 44.

We have inserted the citation as indicated.

- I find that the recent preprint arXiv.2405.02013 beautifully complements this work when thinking about ASQ applications. The authors could consider adding a reference to this preprint in their discussion on the benefits of Ge as an ASQ platform.

We have appended a note at the end of the summary:

After submitting the manuscript, a proximitized quantum dot in Germanium was demonstrated, further highlighting the increasing prominence of Ge in the field of hybrid quantum systems.

- I believe there is a typo in the reference in line 342. An alternative to the current reference would be [A. Bruno et al., Appl. Phys. Lett. 106, 182601 (2015)], but the authors might have had a different option in mind.

We thank the referee for having pointed out the mistake. We have now corrected the reference.

- There are multiple symbol inconsistencies that, while not affecting the understanding of the scientific results, are detrimental to the smooth reading of the text. The manuscript would benefit from a more careful use of symbols. Some examples are: V_{gate} and V_{g} are used to refer to the same variable. Same for I_{c} and I_c . Same for E_J and E_j . Same for E_C and E_c . Same for f_d , f_{d} and f_{drive} . Same for f_q , f_{q} and f_{qubit} . Same for Z , Z_0 and R in the Supplement. The symbol Δ is being used to denote two different things throughout the manuscript (the resonator-qubit detuning and the artificial (effective) Ramsey detuning). I suggest using two different symbols.

We thank the referee for bringing these inconsistencies to our attention. We fixed them.

Yours Sincerely,

Oliver Sagi, Alessandro Crippa and Georgios Katsaros (on behalf of all co-authors)

-
- [1] T. Walter, P. Kurpiers, S. Gasparinetti, P. Magnard, A. Potočnik, Y. Salathé, M. Pechal, M. Mondal, M. Oppliger, C. Eichler, and A. Wallraff, Rapid high-fidelity single-shot dispersive readout of superconducting qubits, Phys. Rev. Appl. **7**, 054020 (2017).
 - [2] E. Jeffrey, D. Sank, J. Y. Mutus, T. C. White, J. Kelly, R. Barends, Y. Chen, Z. Chen, B. Chiaro, A. Dunsworth, A. Megrant, P. J. J. O'Malley, C. Neill, P. Roushan, A. Vainsencher, J. Wenner, A. N. Cleland, and J. M. Martinis, Fast accurate state measurement with superconducting qubits, Phys. Rev. Lett. **112**, 190504 (2014).
 - [3] A. Blais, A. L. Grimsmo, S. M. Girvin, and A. Wallraff, Circuit quantum electrodynamics, Rev. Mod. Phys. **93**, 025005 (2021).
 - [4] R. J. Schoelkopf, A. A. Clerk, S. M. Girvin, K. W. Lehnert, and M. H. Devoret, Qubits as spectrometers of quantum noise, in *Quantum Noise in Mesoscopic Physics* (Springer Netherlands, 2003) p. 175–203.
 - [5] A. Dunsworth, A. Megrant, C. Quintana, Z. Chen, R. Barends, B. Burkett, B. Foxen, Y. Chen, B. Chiaro, A. Fowler, R. Graff, E. Jeffrey, J. Kelly, E. Lucero, J. Y. Mutus, M. Neeley, C. Neill, P. Roushan, D. Sank, A. Vainsencher, J. Wenner, T. C. White, and J. M. Martinis, Characterization and reduction of capacitive loss induced by sub-micron Josephson junction fabrication in superconducting qubits, Applied Physics Letters **111**, 022601 (2017), https://pubs.aip.org/aip/apl/article-pdf/doi/10.1063/1.4993577/14502810/022601_1_online.pdf.
 - [6] C. M. Quintana, A. Megrant, Z. Chen, A. Dunsworth, B. Chiaro, R. Barends, B. Campbell, Y. Chen, I.-C. Hoi, E. Jeffrey, J. Kelly, J. Y. Mutus, P. J. J. O'Malley, C. Neill, P. Roushan, D. Sank, A. Vainsencher, J. Wenner, T. C. White, A. N. Cleland, and J. M. Martinis, Characterization and reduction of microfabrication-induced decoherence in superconducting quantum circuits, Applied Physics Letters **105**, 062601 (2014).
 - [7] O. Dial, D. T. McClure, S. Poletto, G. A. Keefe, M. B. Rothwell, J. M. Gambetta, D. W. Abraham, J. M. Chow, and M. Steffen, Bulk and surface loss in superconducting transmon qubits, Superconductor Science and Technology **29**, 044001 (2016).
 - [8] C. R. H. McRae, H. Wang, J. Gao, M. R. Vissers, T. Brecht, A. Dunsworth, D. P. Pappas, and J. Mutus, Materials loss measurements using superconducting microwave resonators, Review of Scientific Instruments **91**, 091101 (2020).
 - [9] L. Casparis, M. R. Connolly, M. Kjaergaard, N. J. Pearson, A. Kringhøj, T. W. Larsen, F. Kuemmeth, T. Wang, C. Thomas, S. Gronin, G. C. Gardner, M. J. Manfra, C. M. Marcus, and K. D. Petersson, Superconducting gatemon qubit based on a proximitized two-dimensional electron gas, Nature Nanotechnology **13**, 915 (2018).
 - [10] J. Lisenfeld, A. Bilmes, A. Megrant, R. Barends, J. Kelly, P. Klimov, G. Weiss, J. M. Martinis, and A. V. Ustinov, Electric field spectroscopy of material defects in transmon qubits, npj Quantum Information **5**, 105 (2019).
 - [11] J. Wenner, Y. Yin, E. Lucero, R. Barends, Y. Chen, B. Chiaro, J. Kelly, M. Lenander, M. Mariantoni, A. Megrant, C. Neill, P. J. J. O'Malley, D. Sank, A. Vainsencher, H. Wang, T. C. White, A. N. Cleland, and J. M. Martinis, Excitation of superconducting qubits from hot nonequilibrium quasiparticles, Phys. Rev. Lett. **110**, 150502 (2013).